



# Monthly trends of methane emissions in Los Angeles from 2011 to 2015 inferred by CLARS-FTS observations

Kam W. Wong[1,2], Thomas J. Pongetti[1], Tom Oda[3,4], Preeti Rao[1], Kevin. R. Gurney[5], Sally Newman[2], Riley M. Duren[1], Charles E. Miller[1], Yuk L. Yung[2] and Stanley P. Sander[1]

[1]NASA Jet Propulsion Laboratory, California Institute of Technology, Pasadena, California, USA

[2]Division of Geological and Planetary Sciences, California Institute of Technology, Pasadena, California, USA

[3]Goddard Earth Sciences Technology and Research, Universities Space Research Association, Columbia, Maryland, USA

[4]Global Modeling and Assimilation Office, NASA Goddard Space Flight Center, Greenbelt, Maryland, USA

[5]School of Life Sciences, Arizona State University, Tempe, Arizona, USA

*Correspondence to*: K. W. Wong (clare.wong@jpl.nasa.gov)



**Abstract.** This paper presents an analysis of methane emissions from the Los Angeles basin at
monthly timescales across a four-year time period – from September 2011 to August 2015.
Using observations acquired by a ground-based near-infrared remote sensing instrument on
Mount Wilson, California combined with atmospheric $CH_4$-$CO_2$ tracer-tracer correlations, we
observed -18% to +22% monthly variability in $CH_4$:$CO_2$ from the annual mean in the Los
Angeles basin. Top-down estimates of methane emissions for the basin also exhibit significant
monthly variability (-19% to +31% from annual mean and a maximum month-to-month change
of 47%). During this period, methane emissions consistently peaked in the late summer/early fall
and winter. The estimated annual methane emissions did not show a statistically significant trend
over the 2011 to 2015 time period.



## 1 Introduction

Methane ($CH_4$) is a potent and newly regulated greenhouse gas in California. However, its emissions are poorly understood. In the South Coast Air Basin, which holds more than 43% of state's population, the annual methane emissions estimates based on atmospheric $CH_4$ observations indicate that the bottom-up emission inventory was systematically underestimated by 30% to >100% (Wong et al., 2015; Jeong et al., 2013; Peischl et al., 2013; Wennberg et al., 2012; Wunch et al., 2009; Wecht et al., 2014; Cui et al., 2015). Methane sources in the basin can be classified into two categories – biogenic and thermogenic. Biogenic methane is emitted from anaerobic digestion of organic matter by bacteria in waste management facilities, and by cattle in dairy farms. Waste management facilities include landfills, wastewater treatment plants and manure management facilities in dairy farms. Thermogenic methane emissions include natural sources, such as seeps and tar pits, and anthropogenic sources such as natural gas system leakage and gas/oil fields. Emissions from these sources are likely to have different seasonal patterns. Quantifying and tracking the seasonal variability will help us understand methane emissions and are essential for verifying emissions regulation and mitigation policies. However, most studies to date have been based on data from short-term measurement campaigns and have provided limited information on the temporal variability or trends of methane emissions in the basin (Peischl et al., 2013; Wecht et al., 2014; Cui et al., 2015; Wunch et al., 2009).

One commonly used approach to estimate $CH_4$ emissions from atmospheric observations is the tracer-tracer correlation technique. This method uses the regression slopes between observed trace gas mixing ratios (e.g. $CH_4$:$CO_2$ or $CH_4$:$CO$) in the atmosphere to calculate $CH_4$ emissions based on the more accurately known emissions of the correlate (e.g. $CO_2$ or $CO$). This method permits the derivation of the relative emissions of the two trace gases without the use of transport models and does not require the sources to be co-located (Wong et al., 2015; Peischl et al., 2013; Wennberg et al., 2012; Hsu et al., 2010; Wunch et al., 2009). Based on in situ flask observations on Mount Wilson, Hsu et al. (2010) did not observe any seasonal variability in the $CH_4$:$CO$ ratio from April 2007 to February 2008. Using column observations from the Total Carbon Column Observing Network (TCCON) in Pasadena, Wennberg et al. (2012) observed a ±15% monthly variability in the $CH_4$:$CO$ ratio between August 2007 to June 2008, but the monthly variability in methane emissions was not reported.





This paper presents the first study to quantify total methane emissions from an urban region at
the monthly intervals and for an extended period of four years – from September 2011 to August
2015. Using a unique dataset of mountaintop remote sensing observations acquired with the
California Laboratory of Atmospheric Remote Sensing Fourier transform spectrometer (CLARS-
FTS) (Wong et al., 2015; Fu et al., 2014), we have constructed a series of monthly $CH_4$:$CO_2$
tracer-tracer correlations to, address the following questions:
1.  What is the monthly variability in methane emissions in the Los Angeles basin?
2.  Is there a detectable year-to-year methane emissions change in the basin?
3.  What methane source(s) is(are) responsible for any observed temporal trends?
**2  Methods**
Since September 2011, continuous daytime ground-based remote sensing measurements of $CH_4$
and $CO_2$ have been acquired by a JPL-built Fourier transform spectrometer on Mount Wilson
(Wong et al., 2015; Fu et al., 2014). The California Laboratory of Atmospheric Remote Sensing
(CLARS) is located at an altitude of 1670 m above sea level with a panorama of the Los Angeles
basin (Fig. 1). CLARS-FTS quantifies atmospheric column $CH_4$ and $CO_2$ using reflected sunlight
in the near-infrared region. It operates in two measurement modes: Spectralon Viewing
Observations (SVO) and Los Angeles Basin Surveys (LABS). In the SVO mode, the instrument
quantifies the background tropospheric column $CH_4$ and $CO_2$ above the Los Angeles basin by
measuring reflectance from a Spectralon® plate located at the CLARS site. In the LABS mode,
the instrument samples the basin slant column $CH_4$ and $CO_2$ by measuring the surface reflection
from 28 geographical locations (or reflection points) in the basin (Fig. 1). In each measurement
cycle, we collect one set of LABS measurements and four SVO measurements. There are 5 to 8
measurement cycles per day, depending on the time of the year.
Based on the Beer-Lambert Law, the slant column density (SCD) – the total number of absorbing
molecule per unit area along the sun-Earth-instrument optical path – is retrieved for $CH_4$ at 1.67
μm, $CO_2$ at 1.60 μm, and $O_2$ at 1.27 μm using a modified version of the GFIT algorithm
developed at JPL (Fu et al., 2014; Wunch et al., 2011). The retrieved SCDs of $CH_4$ and $CO_2$ are
then converted to slant column-averaged dry air mixing ratio, $XCH_4$ and $XCO_2$, by normalizing



to the retrieved SCD of $O_2$ ($SCD_{O_2}$) (Eq. 1).
$$XGHG = \frac{SCD_{GHG}}{SCD_{O_2}} \times 0.2095 \qquad (1)$$
Individual retrievals are analyzed with multiple post-processing filters to ensure data quality.
Spectra are removed when the residual root mean square errors of the fits to the GFIT radiative
transfer model exceed a pre-defined threshold. These are usually associated with aerosols, high
and low clouds, electrical or mechanical noise, and other transient behavior. Details about the
CLARS-FTS design, data retrieval algorithm and data filtering process are described in Fu et al.
(2014) and Wong et al. (2015).
Wong et al. (2015) mapped the spatial distribution of the $CH_4$:$CO_2$ ratio and derived an annual
total $CH_4$ emission for the basin, based on CLARS-FTS observations from 2011 to 2013. Here
we used the same approach but focused on the temporal trend and quantify the monthly total $CH_4$
emissions for the basin. Therefore, following Wong et al. (2015), we calculated the excess $XCH_4$
and $XCO_2$, due to the emissions from the basin, by subtracting the corresponding SVO
measurements from the LABS observations (Eq. 2).
$$XGHG_{XS} = XGHG_{LABS} - XGHG_{SVO} \qquad (2)$$
We then performed orthogonal distance regression (ODR) analyses of $XCH_{4(XS)}$ and $XCO_{2(XS)}$ for
the 28 reflection points for each month starting from September 2011 to August 2015. To
explore the overall monthly variability during this period, we calculated the weighted average
regression slope among the 28 reflection points, R, using Eq. (3). In Eq. (3), $r_i$ stands for the
regression slope for reflection point i, $w_i$ is the weight which is defined as the reciprocal of the
square of the one sigma uncertainty of the regression slope, $\sigma_i$.
$$R\Big|_{monthly}^{CLARS} = \frac{\sum_{i=1}^{i=28} r_i w_i}{\sum_{i=1}^{i=28} w_i} \text{ , where } w_i = \frac{1}{\sigma_i^2} \qquad (3)$$
**3 Results**



In this section, we describe the monthly and multi-year trends of the basin average regression
slope observed by CLARS-FTS. Figure 2 shows the time series of the Los Angeles basin
weighted average monthly $XCH_{4(XS)}/XCO_{2(XS)}$ regression slopes, R, and their uncertainties
observed by the CLARS-FTS from September 2011 to May 2015. During this period, R ranged
from 5.4 to 7.7 ppb $CH_4$ (ppm $CO_2$)$^{-1}$ with an overall mean of 6.5 ppb $CH_4$ (ppm $CO_2$)$^{-1}$. This is
consistent with previous atmospheric observations: 7.8±0.8 ppb ppm$^{-1}$ from TCCON in 2007-
2008, 6.7±0.6 ppb ppm$^{-1}$ from ARCTAS in 2008, and 6.7±0.0 ppb ppm$^{-1}$ from CalNex in 2010
(Wunch et al., 2009; Wennberg et al., 2012; Peischl et al., 2013). CLARS-FTS observations
showed significant monthly fluctuations. The monthly variability in the slope was -8% to +5% in
2011, -9% to +22% in 2012, -13% to +11% in 2013, -18% to +11% in 2014 and -8% to +11% in
2015. Monthly variability reported here spans the minimum and maximum deviations from the
annual monthly mean for each year. Monthly variability for 2011 and 2015 was calculated based
on partial annual data (that is, from September to December for 2011 and from January to
August for 2015). In general, we observed peaks in late summer, fall and winter: R exceeded 7
ppb $CH_4$ (ppm $CO_2$)$^{-1}$ in August 2012, December 2012, November 2013, August 2014,
September 2014, November 2014 and August 2015. The smallest values of R were observed in
the spring and early summer. Typically, R dipped below 6 ppb $CH_4$ (ppm $CO_2$)$^{-1}$ in May-June,
2012, June 2013, and March 2013.
Figure 3 compares the year-to-year monthly values of R to the four-year mean values. The
weighted four-year mean values showed maxima in August and September, at 7.0 ppb $CH_4$ (ppm
$CO_2$)$^{-1}$. Minima occurred in March when the weighted monthly mean was 5.8 ppb $CH_4$ (ppm
$CO_2$)$^{-1}$. The fall peak was also observed by TCCON observations in Pasadena from 2007 to 2008
(Wennberg et al., 2012). However, no winter peak was observed in their study. CLARS
observations showed multi-year variability for some months but not others. To better understand
the seasonal year-to-year trends in R, we plotted the yearly trends for fall (September, October
and November), winter (December, January and February), spring (March, April and May) and
summer (June, July and August) in Fig. 4. A 15% increase in R over Los Angeles was observed
in the fall season over the last few years. R increased from 6.2 ppb $CH_4$ (ppm $CO_2$)$^{-1}$ in 2012 to
7.1 ppb $CH_4$ (ppm $CO_2$)$^{-1}$ in 2014. This increasing trend was also observed in summer from 2012
to 2014. However, the summer value decreased again from 2014 to 2015. No year-to-year




change was observed in spring. In winter, there were some year-to-year changes but no obvious
increasing or decreasing trend over the study period. The annual average R value showed no
significant trend and less than 4% year-to-year variability between 2011 and 2015.
For comparison, we also calculated the $CH_4$:$CO_2$ emission ratio based on the bottom-up emission
inventory. California Air Resources Board (CARB) reported statewide total emissions of $CH_4$
and $CO_2$ through 2013 (http://www.arb.ca.gov/app/ghg/2000_2013/ghg_sector.php). For $CO_2$,
statewide emissions were 384, 389 and 387 Tg $CO_2$ per year in 2011, 2012, and 2013
respectively. Following Wong et al. (2015), we downscaled the statewide $CO_2$ emissions by
fractional population (43% of state population) to obtain 165, 167 and 166 Tg $CO_2$ per year in
2011, 2012 and 2013, respectively, for emissions from the South Coast Air Basin. For $CH_4$,
bottom-up emissions of 1629, 1636 and 1644 Gg $CH_4$ per year were reported by CARB in 2011,
2012 and 2013 respectively. Following the approach used by Wong et al. (2015), we estimated
the emissions from the South Coast Air Basin by subtracting the agriculture and forestry
emissions from the total emissions and then apportioning the emissions by population. This gave
us emissions of 301, 297 and 300 Gg $CH_4$ per year in the South Coast Air Basin from 2011 to
2013. The bottom-up estimate of R, the $CH_4$/$CO_2$ emission ratio, was calculated from Eq. (4),
where $E_{CH_4}|_{annual}^{inventory}$ is the downscaled CARB annual total $CH_4$ emissions, $E_{CO_2}|_{annual}^{inventory}$ is the
downscaled CARB annual total $CO_2$ emissions and $\frac{MW_{CH_4}}{MW_{CO_2}}$ is the ratio of the molecular weights
of $CH_4$ and $CO_2$ (that is $\frac{16\ g\ CH_4/\ mole}{44\ g\ CO_2/\ mole}$).
$$R_{annual}^{inventory} = \frac{E_{CH_4}|_{annual}^{inventory}}{E_{CO_2}|_{annual}^{inventory}} \times \frac{MW_{CO2}}{MW_{CH4}} \quad (4)$$
Using the downscaled CARB emission estimates for the South Coast Air Basin yields annual R
values of 5.0, 4.9 and 5.0 ppb $CH_4$ (ppm $CO_2$)$^{-1}$ for 2011, 2012 and 2013, respectively. Figure 4
shows that the annual R values determined from CLARS observations are typically in the 6.3 –
6.7 range. Thus, the inventory-based R value systematically underestimated the observed annual
R values by ~30%.





## 1  4 Discussion

We can rearrange Eq. (4) to estimate monthly $CH_4$ emissions from the South Coast Air Basin
using the $CH_4/CO_2$ regression slope R determined from CLARS observations and an inventory-
based estimate of monthly $CO_2$ emissions (Wong et al., 2015).
$$E_{CH_4}|_{monthly}^{top-down} = R|_{monthly}^{CLARS} \times E_{CO_2}|_{monthly}^{inventory} \times \frac{MW_{CH_4}}{MW_{CO_2}} \qquad (5)$$
However, this requires estimates of the monthly $CO_2$ emissions from the South Coast Air Basin.

## 7  4.1 Estimating Monthly $CO_2$ emissions

This subsection explores the available $CO_2$ emission database $(E_{CO_2}|_{monthly})$ for the basin.
CARB reported annual bottom-up statewide $CO_2$ emissions from 2011 to 2013. As described in
the results section, we estimated the annual emissions in the South Coast Air Basin by
apportioning the statewide emissions using the ratio of population in the South Coast Air Basin
to the state population. Because there is no monthly statewide emissions information available,
we distributed the annual $CO_2$ emission evenly over twelve months (shown as solid light blue
line in Fig. 5). Data in 2014 and 2015 (shown as light blue line) are extrapolated using statewide
annual fuel consumption data provided by the Energy Information Administration
(http://www.eia.gov/dnav/ng/hist/n9140us2M.htm;
http://www.eia.gov/dnav/pet/hist/LeafHandler.ashx?n=PET&s=A103450061&f=M ).
In addition to the official CARB emission inventory, three $CO_2$ emission data products provide
monthly temporal resolution for the South Coast Air Basin for our observational period.
1.  Hestia – The Hestia fossil fuel $CO_2$ emissions data product provides sectoral bottom-up
emissions at the building and street level on hourly timescales (http://hestia.project.asu.edu).
Data are available for the South Coast Air Basin for the years 2011 and 2012. Here, we
calculated the monthly total $CO_2$ emissions for the South Coast Air Basin domain based on
the Hestia 1.3 km x 1.3 km hourly gridded version 1.0 (shown by the solid black line in Fig.
5). We defined the South Coast Air Basin domain as the rectangular box bounded by 118.83°
W, 116.67° W, 33.38°N and 34.77°N. Because there are no data after 2012, we extrapolated





1    the emissions from 2012 to 2015 (shown as a faded black line in Fig. 5) using the same
2    approach described above.

3    2.  ODIAC – Open-source Data Inventory for Anthropogenic $CO_2$ (ODIAC) provides global
4        emission fields of fossil fuel $CO_2$ emission with 1 km × 1 km spatial sampling on a monthly
5        basis. ODIAC is based on $CO_2$ emission estimates from the Carbon Dioxide Information and
6        Analysis Center (CDIAC), fuel consumption statistics published by British Petroleum,
7        satellite-observed nightlights and a global power plant database (Oda and Maksyutov, 2011).
8        The monthly $CO_2$ emissions for the South Coast Air Basin domain from September 2011 to
9        December 2014 are shown as the solid red line in Fig. 5. Data in 2015 (shown as the faded
10       red line) are projected using the same approach used to extrapolate the Hestia emissions.

11   3.  FFDAS - Fossil Fuel Data Assimilation System (FFDAS) provides global monthly/hourly
12       sectoral fossil fuel $CO_2$ emission with 0.1° × 0.1° (approx. 10 km × 10 km) spatial sampling
13       (Asefi-Najafabady et al., 2014). This data product is derived from an optimization of the
14       Kaya identity constrained by national fossil fuel $CO_2$ emissions from the International
15       Energy Agency, satellite-observed nightlights, population, and the Ventus power plant
16       dataset. Emissions are available through 2012 (shown as the solid green line). Data from
17       2013 and onwards (shown as the faded green line) are extrapolated using the same method
18       described previously for CARB, Hestia and ODIAC.

As shown in Fig. 5, there are differences as large as 3 Tg $CO_2$ per month among the three
gridded datasets: Hestia, ODIAC and FFDAS. The differences result from 1) emission
calculation methods, 2) the underlying dataset used in the emission calculations and, 3) spatial
modeling. Hestia is derived primarily from local data in the South Coast Air Basin while ODIAC
and FFDAS are based primarily on national and global proxy approaches. It has been shown that
the use of a global dataset may underestimate emissions in Los Angeles by up to 18% (Brioude
et al., 2013). Despite the systematic differences, all three gridded emission datasets show very
similar monthly variability, with peaks in summer and winter. Based on the source
apportionment in Hestia, the summer peak is due to electricity usage (air conditioning) and the
winter peak is due to space heating. In all three datasets, fossil fuel $CO_2$ emissions in the basin
show -9 to +14% monthly fluctuations about the annual mean.



We believe the Hestia data product provide the most accurate $CO_2$ emission estimates for the
South Coast Air Basin among all available databases. Therefore, we used the Hestia $CO_2$
emissions in our calculations to estimate $CH_4$ emissions.
**4.2 Deriving top-down monthly $CH_4$ emissions**
This subsection explains the monthly and annual trends of our methane emission estimates.
Figure 6 shows the time series of monthly methane emissions computed from Eq. (5). Shaded
areas represent the $1\sigma$ uncertainties of the derived emissions. Uncertainties are propagated from
the uncertainties of CLARS-FTS $XCH_{4(XS)}/XCO_{2(XS)}$ regression slopes and $CO_2$ emissions. For
$CO_2$ emissions, we assumed a 10% uncertainty in the Hestia monthly $CO_2$ emissions (K. Gurney,
personal communication, 2016).
Derived methane emission estimates ranged from 23 to 39 Gg $CH_4$ per month. Methane emission
peaks occurred in late summer/early fall and winter months. Distinct peaks of methane emission
occurred in December 2011, August 2012 and December 2012 when methane emissions
exceeded 33 Gg per month. In 2013 and 2014, the summer and fall peaks were less prominent
than in 2012. Minimum methane emissions occurred in late spring/early summer when emissions
dropped below 27 Gg per month. The monthly variability in methane emissions was -12 to +16%
in 2011, -13% to +31% in 2012, -19% to +14% in 2013, -16% to +17% in 2014 and -14% to
+17% in 2015. Monthly variability reported here is the minimum and maximum percent
difference from the annual average. Note that monthly variability in 2011 and 2015 was
calculated based on partial annual data.
Figure 7 plots the monthly patterns of CLARS-FTS inferred methane emissions for each year.
The inferred methane emission estimates showed a bimodal distribution with peaks during the
winter and the late summer/early fall. The weighted monthly average over this period showed
maxima in January, August and December at 31, 33 and 32 Gg $CH_4$ per month. The weighted
monthly average gradually decreased from January to June when methane emission reached a
minimum of 25 Gg $CH_4$ per month. No statistically significant interannual seasonal variability
was observed.





**4.3 Yearly trends in top-down $CH_4$ emissions**
Figure 8 shows the estimated $CH_4$ annual emissions for the South Coast Air Basin from 2011 to
2015. The annual methane emission derived for the South Coast Air Basin was 345 Gg $CH_4$ per
year in 2011. Derived emission increased to 356 Gg $CH_4$ per year in 2013. Since then, there has
been a decreasing trend reaching 325 Gg $CH_4$ per year in 2015. Due to the large uncertainty
propagated mainly from $CO_2$ emissions, we derived a decreasing trend of -5 ± 4 Gg $CH_4$ per year
with only 25% confidence level.
Figure 9 compares all reported $CH_4$ annual total emission estimates for the South Coast Air
Basin in the past ten years. These estimates were derived based on in situ ground observations
(Hsu et al., 2010), column measurements (Wunch et al., 2009, Wennberg et al., 2012; Wong et
al., 2015) and aircraft measurements (Peischl et al., 2013; Wennberg et al., 2012; Wecht et al.,
2014; Cui et al., 2015) in the Los Angeles basin. Among all the previous studies, only one study
(Wong et al., 2015) estimated methane emissions for the period between 2011 and 2015. Our
estimates for 2011 to 2013 were lower but within uncertainties with the estimates reported by
Wong et al. (2015). The difference in the estimated methane emissions between the present study
and Wong et al. (2015) is due to differences in the $CO_2$ reference emissions used in the
calculations. Hestia $CO_2$ emissions used in the present calculations were lower than the
population-scaled CARB emissions used in Wong et al., 2015. The rest of the studies were based
on methane observations from 2007 to 2010. Despite the different study periods, methane
emission estimates from our study are inconsistent with previous top-down estimates. About half
of previously reported methane emission estimates were focused on the CALNEX field
experiment in May and June 2010. The annual methane emission estimates from these studies
could be underestimated as we observed that methane emissions tend to be lowest during these
months. Comparing our results to the bottom-up inventory, the scaled CARB $CH_4$ emissions
from 2011 to 2013 were 2-31% lower than our estimates.
**4.4 Analysis assumptions**
In this subsection, we discuss the analysis assumptions used to derive $CH_4$ emissions for the
South Coast Air Basin using CLARS-FTS observations.



- **Spatial and temporal representation based on CLARS-FTS measurement technique**. We assumed that the CLARS-FTS measurement domain is representative of the South Coast Air Basin. The CLARS-FTS measurement domain covers 67% of $CO_2$ emissions in the South Coast Air Basin spatial domain according to the Hestia $CO_2$ data product. Therefore, the CLARS-FTS observations are more representative of the sampled area in the South Coast Air Basin than the entire basin. In addition, our methane emission estimates were based on daytime-only observations.

- **Spatial and temporal bias due to data filtering.** CLARS-FTS samples the Los Angeles basin using its standard measurement sequence. However, as described in Wong et al. (2015), certain months of the year are more prone to cloud and aerosol interference in the Los Angeles basin. This may introduce biases in the monthly sampling of post-filtered data. To accurately estimate the LA basin value, we used the weighted average $XCH_{4(XS)}/XCO_{2(XS)}$ regression slope, as the statistical weight for each reflection point is based on the number of samples passing through the data quality filters. We also performed a bootstrap analysis to ensure that there is no sampling bias in the regression slopes (Efron and Tibshirani, 1993).

- **Seasonal bias due to transport variability.** Changes in meteorology patterns in summer vs. winter can lead to a seasonal dependence on the observations' footprint, which is the sensitivity of the observations to changes in emissions. In the Los Angeles basin, the prevailing winds are typically northwesterly and onshore throughout the year, except for Santa Ana events (Conil and Hall, 2006). During Santa Ana events, which typically occur during the period from October to March, the wind patterns in the basin shift to easterly and offshore flow (Hughes and Hall, 2010). We investigated the impact of Santa Ana events on our results using the Santa Ana Index to remove observations during Santa Ana events (Hughes and Hall, 2010; Conil and Hall, 2006; http://meteora.ucsd.edu/weather/). A correlation analysis showed that applying the Santa Ana Index filter did not cause any statistically significant bias on the CLARS monthly $CH_4$:$CO_2$ ratios. This insensitivity is likely due to the effect of spatial averaging over 28 slant column measurements that span a 50 x 100 $km^2$ spatial domain in the Los Angeles basin, mitigating the effect of transport variability, especially when compared with measurements from individual tower sites. A more diagnostic approach involving the application of a high-resolution tracer transport



model to investigate potential transport-induced biases on CLARS-FTS results will be
carried out in the future.

### 4.5 Exploring seasonal variability from major CH$_4$ emission sources

Currently, no monthly methane emission database is publicly available for comparison with our
top-down estimates during our observational period. In this subsection of the paper, we review
previous studies of the seasonal emissions variability from major methane sources (landfills,
dairies, wastewater treatment plants and natural gas system leakage) to understand possible
contributions to the observed monthly variability in total CH$_4$ emission in the South Coast Air
Basin.
• **Landfills.** Landfills are major emitters of CH$_4$ in the basin. Previous studies suggested that
landfills could contribute 41-63% of total annual methane emissions (Peischl et al., 2013;
Wennberg et al., 2012; Hsu et al., 2010). The seasonal variability in landfill CH$_4$ emissions is
poorly understood, however. Peischl et al. (2013) estimated the emissions from two of the
largest landfills in the basin – Olinda Alpha landfill and Puente Hills landfill – based on
aircraft measurements in May and June 2010. Based on observations taken from four flights
in May and one flight in June, their studies found that CH$_4$ emissions from Olinda Alpha
landfill was almost double in June relative to May while Puente Hills landfill (which was
closed in 2012) showed less than 15% changes in monthly emissions in 2010. Using a
landfill model, Spokas et al. (2015) found that the statewide landfill emissions were largest in
October and smallest in April in 2010. Other observational studies found that CH$_4$ emissions
from landfills peak in July and August (Shan et al., 2013; Spokas et al., 2011; Tratt et al.,
2014; Goldsmith et al., 2012). These studies suggest that landfills can contribute to the late
summer/early fall peak in the total CH$_4$ emissions observed by CLARS-FTS but are unlikely
to explain the winter peaks.
• **Dairies.** Previous observations suggested that dairy farms could contribute 32 – 76 Gg CH$_4$
per year in the South Coast Air Basin (Peischl et al., 2013; Wennberg et al., 2012). This
corresponds to 8% to 36% of the reported total annual CH$_4$ emissions in the studies. In
general, studies on dairies focus on mitigation strategies rather than quantifying temporal
changes in emissions. Limited studies of dairy emissions report peaks in CH$_4$ emissions in





summer and early fall (from June to September), and steady minima in spring and winter
(VanderZaag et al., 2014; VanderZaag et al., 2013; VanderZaag et al., 2010; VanderZaag et
al., 2009; Ulyatt et al., 2002; Kaharabata et al., 1998). These findings imply that dairies can
also be contributing to the summer/early fall peaks in the CLARS-FTS inferred $CH_4$
emissions.
• **Wastewater treatment**. This sector is shown to be responsible for 33% of Los Angeles
County and 9.4% of the South Coast Air Basin (Hsu et al., 2010; Wennberg et al., 2012).
Daelman et al. (2012; 2013) measured $CH_4$ emissions from a wastewater treatment facility
for one year (2010-2011) and reported up to 40% monthly fluctuations from the mean, with a
maximum in June.
• **Fossil fuel sources.** Recent studies based on mobile, stationary and airborne measurements
of methane in Los Angeles indicated that fossil fuel sources contribute 47% to 90% of the
total $CH_4$ emissions in the basin (Wennberg et al., 2012; Townsend-Small et al. 2012; Peischl
et al., 2013; Hopkins et al., 2015). Wennberg et al. (2012) and Peischl et al. (2013) suggested
that fugitive emission from natural gas distribution system leakage contributes to the gaps
between bottom-up and top-down total $CH_4$ emissions in the South Coast Air Basin. McKain
et al. (2014) found little seasonal dependence (<10%) on the emissions from the natural gas
system in Boston, Massachusetts. Their studies showed a leakage rate of 2.7 ± 0.6% from the
natural gas system. Wennberg et al. (2012) reported a consistent leakage rate from the natural
gas system in Los Angeles and suggested that most of the leakages from such systems are
likely to occur in residential/commercial areas where the distribution system ends. Publicly
available natural gas consumption data from residential and commercial sectors in the South
Coast Air Basin show a significant seasonal cycle with a maximum in winter due to heating
(https://energydatarequest.socalgas.com/). Wennberg et al. (2012) and McKain et al. (2014)
observed that the leakage rate from the natural gas system is constant throughout the year and
suggested that the majority of leakage occurs in the distribution system to the residential and
commercial sectors. This conclusion is reasonable since the natural gas distribution pipeline
system is pressure-regulated at several points, and leakage should be independent of
consumption to first order. However, this is not the case for natural gas storage facilities
which are pressurized to higher levels in the summer and late fall in Southern California to
respond to increased demands for summertime electric power generation for air conditioning





and wintertime space heating. In October, 2015, a massive leak began at an underground well
pipe at the Aliso Canyon (Los Angeles) natural gas storage facility as it was being
pressurized to provide wintertime reserves. While this leak was unprecedented in scale, it
raises the question whether smaller fugitive leaks in the storage infrastructure from this and
numerous other above- and below-ground reservoirs contribute to the seasonal variability
observed in CLARS-FTS data. The Aliso Canyon leak resulted in very large increases (as
much as a factor of 10) in the observed instantaneous values of $XCH_{4(XS)}/XCO_{2(XS)}$ throughout
the entire CLARS-FTS field of regard (Wong et al., in prep.). Since CLARS-FTS is capable
of resolving $CH_4$ enhancements that are significantly smaller than those caused by the Aliso
Canyon leak, perhaps seasonally-varying fugitive emissions from natural gas storage
facilities and associated infrastructure are partially responsible for the observed monthly
variability. Enhanced long-term monitoring for fugitive emissions will be required to test this
hypothesis.
**5 Summary and Conclusions**
Using CLARS-FTS mountaintop remote sensing observations from Mount Wilson along with
tracer-tracer $CH_4$:$CO_2$ correlation analyses, we estimated the monthly variability in $CH_4$:$CO_2$ and
top-down $CH_4$ emissions from the South Coast Air Basin from 2011-2015. Significant monthly
variability (-18% to +22%) in $CH_4$:$CO_2$ was observed. Double peaks in late summer/early fall
and winter occurred consistently during the study period. The fall peak in the $CH_4$:$CO_2$ ratios
was also observed by TCCON (Wennberg et al., 2012). The CLARS-FTS $XCH_{4(XS)}/XCO_{2(XS)}$
regression slopes showed -7% to 10% year-to-year seasonal variability, with an increasing trend
in the fall season from 2012 to 2014. The annual average $XCH_{4(XS)}/XCO_{2(XS)}$ regression slopes
showed less than 4% year-to-year variability between 2011 and 2015.
Using the best available estimates of $CO_2$ emissions, top-down estimates of $CH_4$ emissions were
determined using the emission ratio method. Repeatable peaks in late summer/early fall and
winter were observed between 2011 and 2015. There were significant monthly fluctuations (-
19% to +31% from annual mean and a maximum month-to-month change of 47%) in the
inferred methane emissions in the basin. Based on previous studies on the seasonal variability of



$CH_4$ emissions from $CH_4$ sources, we concluded that landfills, dairies and wastewater treatment
facilities are likely sources of the peak $CH_4$ emissions in late summer/early fall. Fugitive
emissions from natural gas storage facilities and associated infrastructure may contribute to both
the late summer and late fall peaks.
No significant trend in $CH_4$ emissions  (-5 ± 4 Gg $CH_4$ per year with a 25% confidence level due
to the uncertainty in $CO_2$ emissions) could be discerned over the 2011 to 2015 time period. The
population-scaled bottom-up $CH_4$ emissions from 2011 to 2013 were 2-31% lower than our top-
down estimates. These results are consistent with previous studies (Wunch et al., 2009; Hsu et
al., 2010; Wennberg et al., 2012; Peischl et al., 2013; Wong et al., 2015). A combination of
several measurement and modeling strategies are necessary to further disentangle the monthly
variability of methane sources in the Los Angeles basin.
**Acknowledgements**
The research in this study was performed at the Jet Propulsion Laboratory, California Institute of
Technology, under a contract with the National Aeronautics and Space Administration. KWW
thanks the California Air Resources Board, NIST GHG and Climate Science Program, and the
W.M. Keck Institute for Space Studies for support. The authors would like to acknowledge our
colleagues at JPL and California Institute of Technology, and Risa Patarasuk at Arizona State
University for helpful comments and suggestions.



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





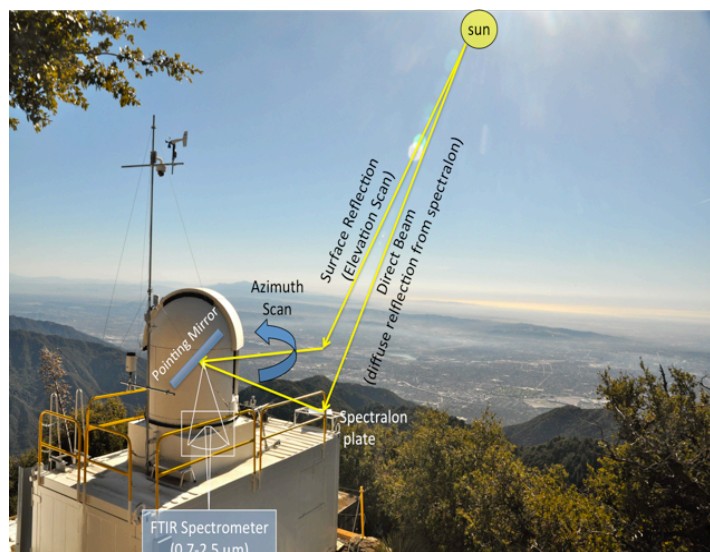

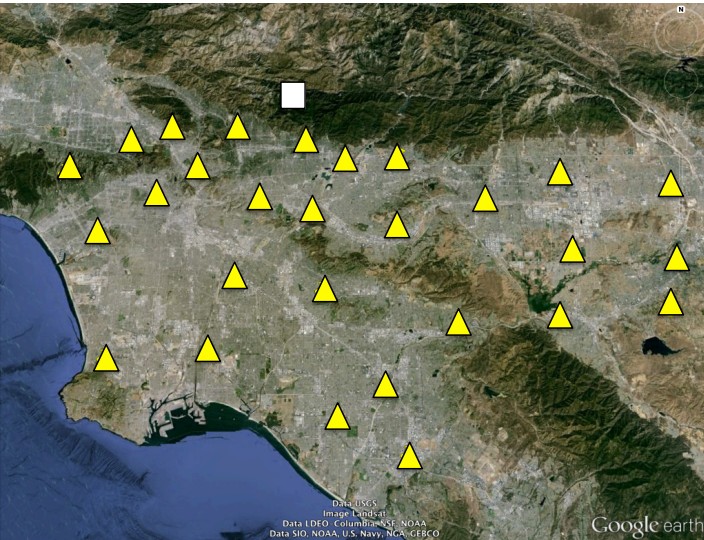

Figure 1. Top: CLARS facility located at 1.67 km above sea level on Mount Wilson, looking
over the Los Angeles basin. Optical paths from direct sun beam and basin surface reflection are
shown as yellow lines. Bottom: Location of 29 reflection points on Mount Wilson (white square)
and in the basin (yellow triangles).





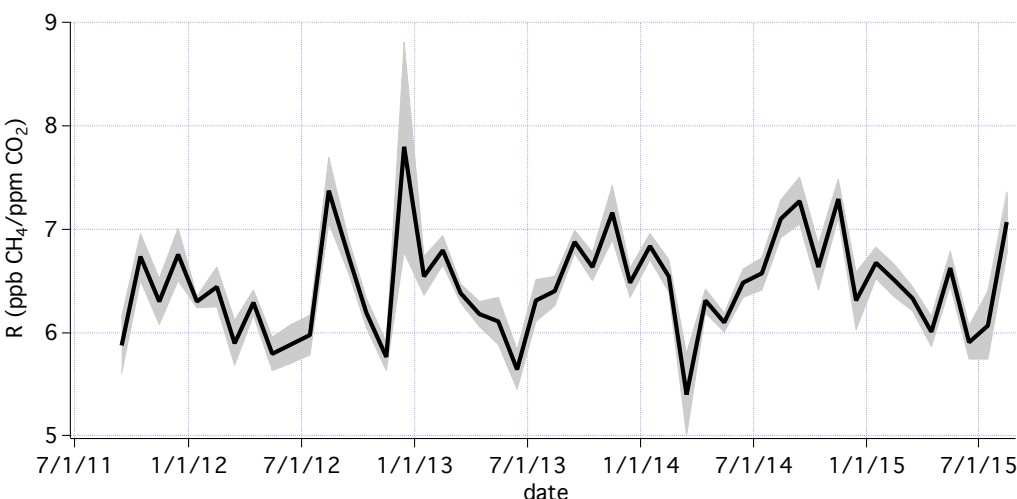

3    Figure 2. Time series of the Los Angeles basin weighted-average monthly regression slopes of

4    $XCH_{4(XS)}$—$XCO_{2(XS)}$ (in unit of ppb $ppm^{-1}$) and their uncertainties observed by the CLARS-FTS

5    in the basin from September 2011 to May 2015. Uncertainties are ±1σ of the regression slopes.





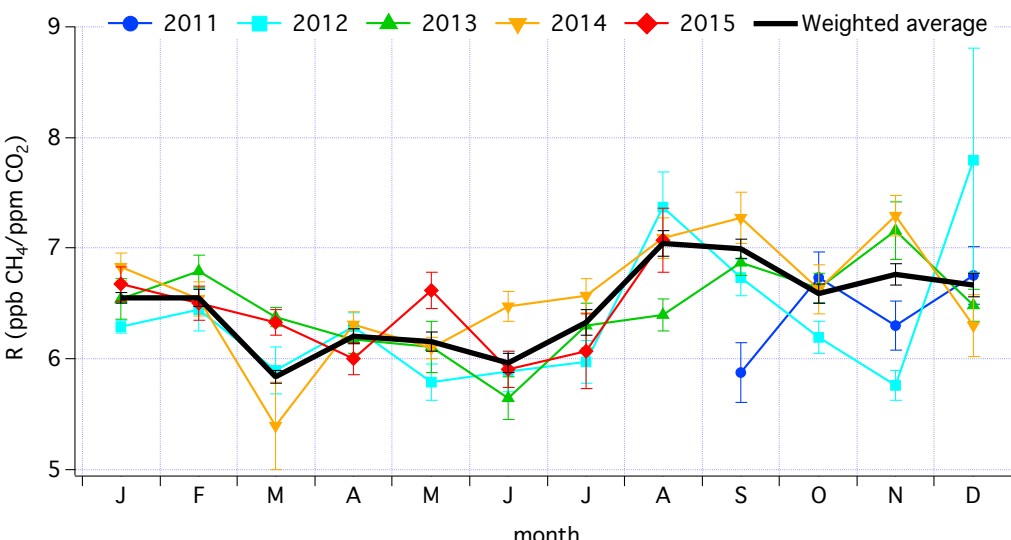

Figure 3. Monthly patterns of the Los Angeles basin weighted-average regression slopes of
$XCH_{4(XS)}$—$XCO_{2(XS)}$ (in unit of ppb ppm$^{-1}$) and their uncertainties observed by the CLARS-FTS
in the basin. Monthly trends are color coded as follows: 2011 in blue, 2012 in cyan, 2013 in
green, 2014 in orange and 2015 in red. Monthly average ratio and its standard deviation over the
entire observational period are shown in black.



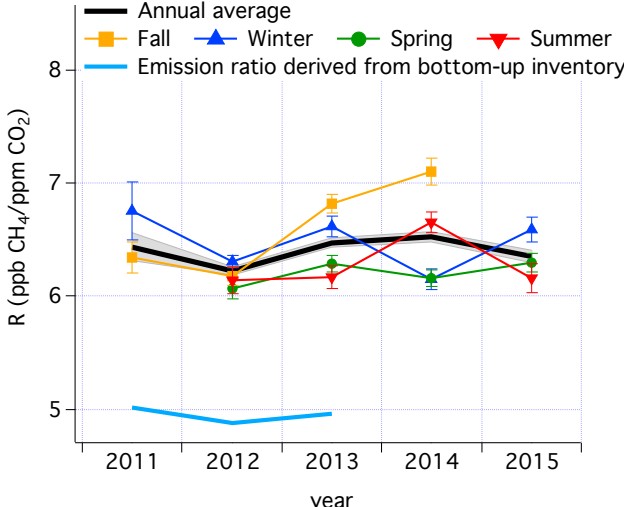

Figure 4. Interannual variability of R (in units of ppb $CH_4$ (ppm $CO_2$)$^{-1}$) in fall (orange), winter
(blue), spring (green) and summer (red) from 2011 to 2015. The annual average ratio is shown in
black. Also shown are the ±1σ uncertainties. Note that data for 2011 and 2015 are derived from
partial annual observations (that is, September to December for 2011 and January to August for
2015. The $CH_4$:$CO_2$ ratio based on the population-scaled bottom-up emission inventory from the
California Resources Board is shown in light blue (California Air Resources Board, 2013).




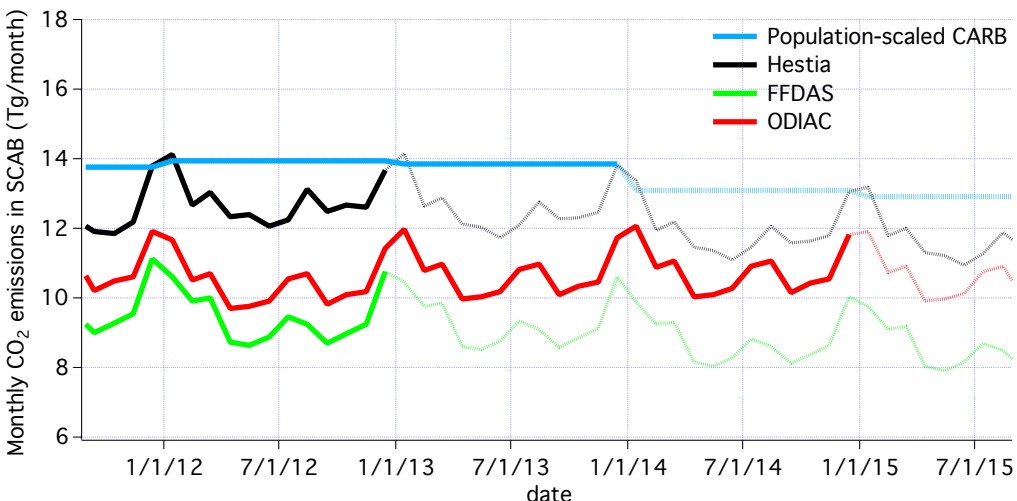

Figure 5. Time series of the different $CO_2$ monthly emissions (in units of Tg per month) from the
South Coast Air Basin. Emissions are color coded as follows: Population-scaled CARB in light
blue, Hestia in solid black, ODIAC in solid red and FFDAS in solid green. Extrapolated
emissions using annual fuel consumption data are shown in faded solid lines.





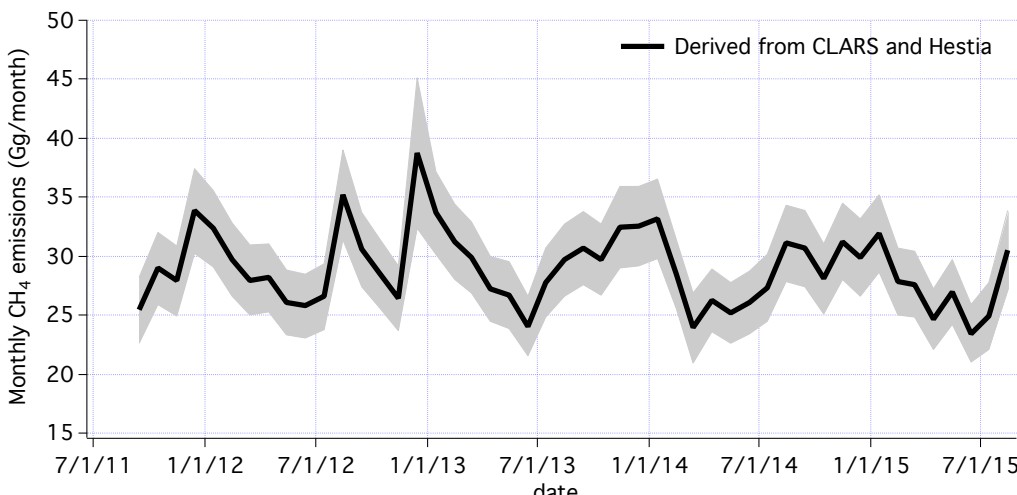

Figure 6. Time series of CLARS-FTS inferred monthly $CH_4$ emissions (in units of Gg per
month) and their 1σ uncertainties from the Los Angeles basin from September 2011 to August
2015. Overall uncertainties are propagated from the uncertainties of CLARS-FTS $XCH_{4(XS)}$—
$XCO_{2(XS)}$ regression slopes and $CO_2$ emissions.





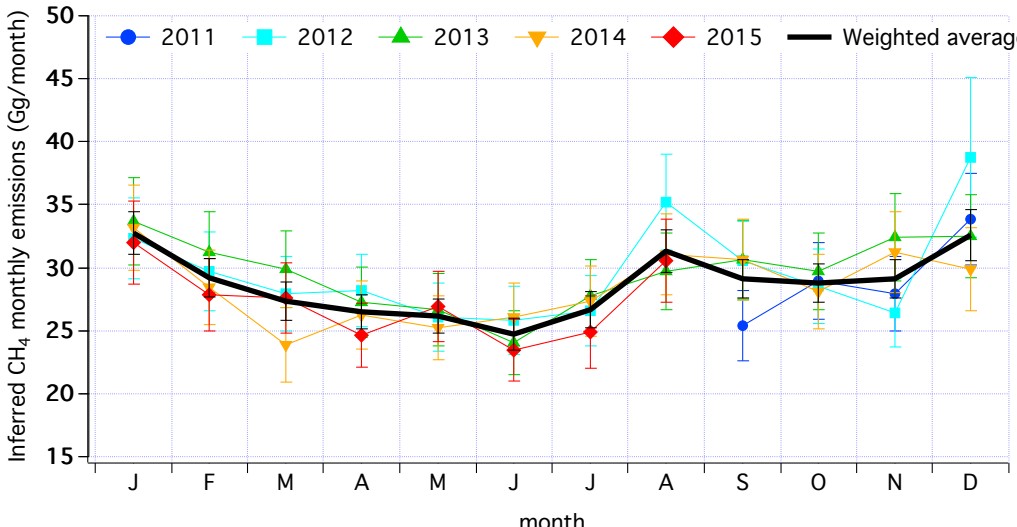

Figure 7. Monthly patterns of derived $CH_4$ emissions (in units of Gg per month). Error bars
represent the $\pm 1\sigma$ uncertainties. Derived $CH_4$ emissions are color coded as follows: 2011 in blue,
2012 in cyan, 2013 in green, 2014 in orange and 2015 in red. Average monthly emissions and
their standard deviations over the entire observational period are shown in black.



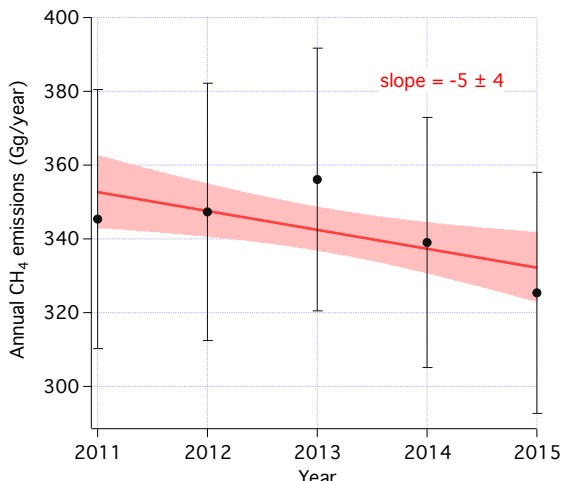

3    Figure 8. CLARS-FTS inferred annual CH$_4$ emission estimates (in units of Gg per month), based

4    on Hestia CO$_2$ emissions. Red line indicates the regression slope and the shaded area is the 25%

5    confidence interval.





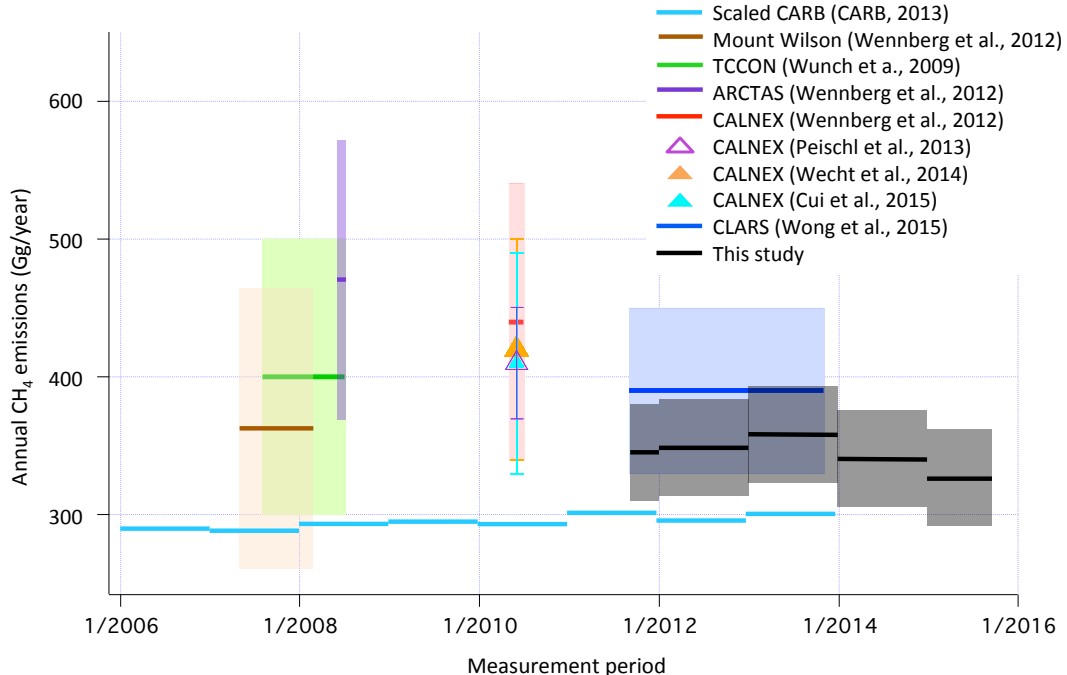

2 Figure 9. Comparison of annual CH$_4$ emission estimates (in unit of Gg per month) reported in the

3 past ten years. The Mount Wilson estimate reported by Wennberg et al. (2012) was derived for

4 the South Coast Air Basin using the emission estimates based on Hsu et al., 2012.