# Peer review of "Monthly trends of methane emissions in Los Angeles from 2011"

_Atmospheric Chemistry and Physics, 2016_

## Referee Comment (RC1) · Anonymous Referee #1 · 2 Jun 2016

General Comments:

Wong et al. report measurements of $CO_2$ and $CH_4$ between 2011 and 2015 from a remote sensing instrument located on Mt. Wilson near Los Angeles, California, USA. Column $CO_2$ and $CH_4$ measurements above the instrument are subtracted from measurements from 28 points around the Los Angeles Basin in order to obtain an excess column enhancement below the instrument. These enhancements are fit linearly to provide excess $CH_4$ to $CO_2$ ratios, which are compared on a monthly basis to the other monthly measurements, and to previous studies. The ratios are also multiplied by $CO_2$ emission estimates from the basin to provide a monthly $CH_4$ emission. $CH_4$ emissions generally peaked in the late summer/early fall and wintertime in the Los

Angeles Basin.

Overall, this paper is well-written and needs only minor revisions. However, the conclusions are seemingly at odds with another recent ACPD submission, Wunch et al. Whereas Wong et al. find Los Angeles Basin CH4 emissions decreasing over the 2011 to 2015 time period, Wunch et al. report increasing CH4 emissions from 2012 to 2015. I therefore suggest some discussion of the Wunch et al. results.

Specific Comments:

Although there are many references to Wong et al. (2015), an example correlation plot of XCH4:XCO2 would be useful in the current work.

p.14, line 7, the authors state Hsu et al. or Wennberg et al. showed wastewater treatment was responsible for the emissions stated. Both reported inventory values for wastewater treatment, but could not verify those inventories were correct. Since the definition of "showed" could be either "proved" or just simply "presented", the authors should clarify this statement.

Figure 5, I am curious what it would look like if average daily emissions were shown per month. As presented, the months with 31 days always seem to have small peaks compared to the surrounding months with less than 31 days.

Technical Comments:

p.4, line 4, capitalize Transform Spectrometer p.4, line 6, remove comma between "to address" p.4, line 26, change to "molecules" p.7, line 18, change ratio to MWCO2/MWCH4 to match equation p.8, is there a peer-reviewed citation for Hestia? p.9, line 21, move comma to read "calculations, and (3) ..." References, add Peischl et al. Figure 6, line 5, suggest changing dash to colon for CH4:CO2 ratio

---

## Referee Comment (RC2) · Anonymous Referee #2 · 28 Jul 2016

Report of referee #2

General comments :

This paper combines column ground-based measurements of CH4 and CO2 and tracer-tracer correlation technique to provide monthly estimates of CH4 :CO2 ratios (R) in the Los Angeles basin over a four year period (Sept. 2011 – Aug.2015). Methane emissions are then estimated by combining the R estimates with CO2 emissions from bottom-up emission inventories. Some efforts have been made to take into account for the monthly variability of these emissions into the inventories.

A specific feature of this paper is that it relies on remarquable FTS datasets collected

from the CLARS instrument on Mt Wilson, pointing both above and within the LA basin.

This study was a real pleasure to review. It is very well written and clear, adressing the hot topic of improving urban greenhouse gas emission estimates. It is very well suited for publication in ACP. I have only some minor revisions to advice and a few questions for the authors.

Specific comments :

What were the constrains that led to a number of 28 points for the LABS measurements ? How well do 28 points represent the spatial variability of the CH4 emissions in the LA basin ? By reading your paper further, I see this question is a bit adressed on p.12, but it could be discussed more. And, at least one sentence would be welcome in the methods section (p.4) to explain why you ended-up to this number of 28 sites.

Regarding possible biases relative to advection, would it be technically feasible, and do you think it would be correct, to point the FTS to the surface on background up-wind areas (i.e. not contaminated by LA emissions), and then infer the urban plume (XCH4svo-XCH4bkg) :(XCO2svo-XCO2bkg) ratio rather than the XCH4xs :XCO2xs described in your paper ?

How many observations did you collect per month ? Are there some months with low number of observations (issues with clouds...) ? Can this cause biases in the comparison of the R estimate from one month to the other ? Please better quantify this piece of information. See below my comment on p.10 lines19-20.

Also, do you have the same amount of observations for each hour of the day (do you have biases linked to the hour you were able to collect measurements regarding clear sky conditions) ?

Detailed comments :

p.4 Line 23 : the chosen strategy is, in each measurement cycle, to collect one set of LABS measurements and four SVO measurements. Please explain the motivation for

this strategy.

p.6 lines 5-7 : please choose one single notation (ppm CO2)-1 or ppm-1

p.6 lines 6-7 : please make it clearer : are the +/-0.8 and +/-0.0 indicating the uncertainty of the results or their variability ? How do the uncertainties compare between this study and the former ones ?

p.7 line 30 : The inventory-R based value underestimation of 30% seems effectively much larger than the CLARS R uncertainties, but please quantify this later (apparently from Fig.2, something like 3% ?).

p.8 lines 12-13 : You made the choice of distributing regularly the CARB CO2 emissions on the twelve months of the year. However, as you mention p.9 lines 25-26, the three better resolved inventory show similar monthly variability. Why don't you apply this variability around the mensual mean to distribute the annual CARB CO2 emissions ? This would likely be more accurate and interesting to compare with the three highly resolved inventories.

p.10 line 1 : Please explain why you believe more in Hestia estimates than in the others.

p.10 lines 9-10 : K. Gurney is a co-author of the study, please remove (K. Gurney, 10 personal communication, 2016).

p.10 lines 19-20 : please quantify what Âń partial Âż means here (see my specific comments).

p.13-15 : It would be interesting to give also here some information on the role of the different emission sectors as seen by the monthly-resolved inventories, and to compare this information with the top-down results cited in this section.

Please also note the supplement to this comment:
http://www.atmos-chem-phys-discuss.net/acp-2016-232/acp-2016-232-RC2-supplement.pdf

---

## Author Comment (AC1) · 6 Sep 2016

**Response to Anonymous Referee #1**

We would like to thank the reviewer for taking time to review our paper and provide us with helpful comments. Please find our response to the reviewer's comments in blue in the following.

**General comments:**

Wong et al. report measurements of CO2 and CH4 between 2011 and 2015 from a remote sensing instrument located on Mt. Wilson near Los Angeles, California, USA. Column CO2 and CH4 measurements above the instrument are subtracted from measurements from 28 points around the Los Angeles Basin in order to obtain an excess column enhancement below the instrument. These enhancements are fit linearly to provide excess CH4 to CO2 ratios, which are compared on a monthly basis to the other monthly measurements, and to previous studies. The ratios are also multiplied by CO2 emission estimates from the basin to provide a monthly CH4 emission. CH4 emissions generally peaked in the late summer/early fall and wintertime in the Los Angeles Basin.

Overall, this paper is well-written and needs only minor revisions. However, the conclusions are seemingly at odds with another recent ACPD submission, Wunch et al. Whereas Wong et al. find Los Angeles Basin CH4 emissions decreasing over the 2011 to 2015 time period, Wunch et al. report increasing CH4 emissions from 2012 to 2015. I therefore suggest some discussion of the Wunch et al. results.

Response: Thank you for the reviewer's nice comments. Regarding the trend of annual $CH_4$ emissions over the 2011 to 2015 time period, our study concluded that there was no statistically significant trend in annual total $CH_4$ emissions over the 2011 to 2015 time period. In the abstract, the last sentence stated "The estimated annual methane emissions did not show a statistically significant trend over the 2011 to 2015 time period". Wunch et al. reported annual methane emissions from 380±78 Gg $CH_4$/yr, 352±71 Gg $CH_4$/yr and 448±91 Gg $CH_4$/yr for the period 9/2012—8/2013, 9/2013—8/2014 and 9/2014—8/2015 respectively. It seems that there is not a significant trend in the emissions because the uncertainties overlap. Because Wunch et al. (2016) is still currently under review and has not been published, we did not cite their results in our paper. If their study is published, we would be happy to include a discussion of their results in our paper.

**Specific Comments:**

1. Although there are many references to Wong et al. (2015), an example correlation plot of XCH4:XCO2 would be useful in the current work.

Response: Thanks for the suggestion. We have added the following figure as an example correlation plot of $XCH_4$:$XCO_2$ excess in the supplemental material.

[Figure]

Figure S1. Scatter plot showing an example of correlation between $XCH_4$ excess and $XCO_2$ excess for CLARS-FTS west Pasadena target in January 2015. Regression slope of $7.4\pm0.9$ was observed during this period.

2. p.14, line 7, the authors state Hsu et al. or Wennberg et al. showed wastewater treatment was responsible for the emissions stated. Both reported inventory values for wastewater treatment, but could not verify those inventories were correct. Since the definition of "showed" could be either "proved" or just simply "presented", the authors should clarify this statement.

Response: We have clarified this by changing the word "showed" to "suggested". The sentence (now at line 24 page 14) is updated to "This sector is suggested to be responsible for 33% of Los Angeles County and 9.4% of the South Coast Air Basin (Hsu et al., 2010; Wennberg et al., 2012)".

3. Figure 5, I am curious what it would look like if average daily emissions were shown per month. As presented, the months with 31 days always seem to have small peaks compared to the surrounding months with less than 31 days.

Response: This is a good suggestion. We looked into that and found that the seasonal cycle does not have a significant change. Therefore, we decide to keep the units as total emission per month.

**Technical Comments:**

1. p.4, line 4, capitalize Transform Spectrometer

Response: The words "transform spectrometer" in "Fourier transform spectrometer" should be in lower case. Therefore, we did not make this change in our text.

2. p.4, line 6, remove comma between "to address"

Response: The change has been made in the text.

3. p.4, line 26, change to "molecules"

Response: The change has been made in the text. Please refer to line 2 on page 5 of the revised manuscript.

4. p.7, line 18, change ratio to MWCO2/MWCH4 to match equation

Response: Changes have been made to the text. The sentence (now at line 22 of page 7) is updated to "The bottom-up estimate of R, the $CH_4/CO_2$ emission ratio, was calculated from Eq. (4), where $E_{CH_4}|_{annual}^{inventory}$ is the downscaled CARB annual total $CH_4$ emissions, $E_{CO_2}|_{annual}^{inventory}$ is the downscaled CARB annual total $CO_2$ emissions and $\frac{MW_{CO_2}}{MW_{CH_4}}$ is the ratio of the molecular weights of $CH_4$ and $CO_2$ (that is $\frac{44\ g\ CO_2/\ mole}{16\ g\ CH_4/\ mole}$)."

5. p.8, is there a peer-reviewed citation for Hestia?

Response: No, the Los Angeles Hestia project does not have a peer-reviewed citation yet. The best citation for this project is http://hestia.project.asu.edu, which we have included in the paper already.

6. p.9, line 21, move comma to read "calculations, and (3) . . ."

Response: The comma has been moved. The sentence (now at line 29 of page 9) has been updated to "The differences result from 1) emission calculation methods, 2) the underlying dataset used in the emission calculations, and 3) spatial modeling."

7. References, add Peischl et al.

Response: Changes have made in the text. In the reference section (line 9 on page 20), Peischl et al. 2013 has been added as "Peischl, J., Ryerson, T. B., Brioude, J., Aikin, K. C., Andrews, A. E., Atlas, E., Blake, D., Daube, B. C., de Gouw, J. A., Dlugokencky, E., Frost, G. J., Gentner, D. R., Gilman, J. B., Goldstein, A. H., Harley, R. A., Holloway, J. S., Kofler, J., Kuster, W. C., Lang, P. M., Novelli, P. C., Santoni, G. W., Trainer, M., Wofsy, S. C., and Parrish, D. D.: Quantifying sources of methane using light alkanes in the Los Angeles basin, California, J. Geophys. Res.- Atmos., 118, 4974–4990, doi:10.1002/jgrd.50413, 2013."

8. Figure 6, line 5, suggest changing dash to colon for CH4:CO2 ratio

Response: Thanks for the suggestion. We used the colon in past for our previous publication but the ACP journal typesetting suggested the use of dash instead. Therefore, we decide to keep the dash.

---

## Author Comment (AC2) · 6 Sep 2016

**Response to Anonymous Referee #2**

We would like to thank the reviewer for providing us with helpful comments. Please find our response to the reviewer's comments in blue in the following.

**General comments:**

This paper combines column ground-based measurements of CH4 and CO2 and tracer-tracer correlation technique to provide monthly estimates of CH4 :CO2 ratios (R) in the Los Angeles basin over a four year period (Sept. 2011 – Aug.2015). Methane emissions are then estimated by combining the R estimates with CO2 emissions from bottom-up emission inventories. Some efforts have been made to take into account for the monthly variability of these emissions into the inventories.

A specific feature of this paper is that it relies on remarquable FTS datasets collected from the CLARS instrument on Mt Wilson, pointing both above and within the LA basin.

This study was a real pleasure to review. It is very well written and clear, adressing the hot topic of improving urban greenhouse gas emission estimates. It is very well suited for publication in ACP. I have only some minor revisions to advice and a few questions for the authors.

Response: We are glad that the reviewer enjoyed reading this paper. Thank you for the nice comments. We have made changes and edits in the text to address the comments from the reviewer.

**Specific Comments:**

1. What were the constrains that led to a number of 28 points for the LABS measurements? How well do 28 points represent the spatial variability of the CH4 emissions in the LA basin? By reading your paper further, I see this question is a bit addressed on p.12, but it could be discussed more. And, at least one sentence would be welcome in the methods section (p.4) to explain why you ended-up to this number of 28 sites.

Response: The constrains that led to a number of 28 points for the LABS measurements is a balance of the measurement spatial coverage of the Los Angeles basin and the temporal coverage during the day. Since the CLARS-FTS only operates during daytime, if we increase the number of reflection points, the time taken to perform one measurement cycle will increase. This will lead to a decrease of the number of measurement cycles performed per day or a decrease of temporal information. We found that having 28 reflection points for the LABS measurement will allow us to have optimal spatial and temporal coverage for the basin. The number of reflection points can be easily modified. Indeed, since December 2015, we have expanded our spatial domain to include five more reflection points in the San Fernando Valley to address the Aliso Canyon methane leak. At line 22 on page 4, we added the following sentence "We selected 28 reflection points to achieve

an optimal spatial and temporal coverage of the Los Angeles basin. The number, locations and repeat frequencies of the reflection points can be easily modified to meet specific measurement requirements."

2. Regarding possible biases relative to advection, would it be technically feasible, and do you think it would be correct, to point the FTS to the surface on background up- wind areas (i.e. not contaminated by LA emissions), and then infer the urban plume (XCH4svo-XCH4bkg) :(XCO2svo-XCO2bkg) ratio rather than the XCH4xs :XCO2xs described in your paper ?

Response: We did consider measuring the reflection points upwind and downwind of the city. However, we believe that it is difficult to get a clean background for an upwind location (for example, Santa Monica or Marina Del Rey). This is because of the long optical path in the boundary layer for these locations. We observed the effect of the boundary layer in basin $XCH_4$ and $XCO_2$, but not in the Spectralon $XCH_4$ and $XCO_2$ (Wong et al. 2015). Thus, we believe the Spectralon is the best background in our observations and using $XCH_4(xs):XCO_2(xs)$ to infer the urban plume is a good approach. We have not modified the paper in response to this comment.

3. How many observations did you collect per month ? Are there some months with low number of observations (issues with clouds...) ? Can this cause biases in the comparison of the R estimate from one month to the other ? Please better quantify this piece of information. See below my comment on p.10 lines19-20.

Response: We typically collect about 150-200 observations (pre-filtered) for each reflection points per month with good sky conditions. Some months had less observations due to the weather conditions such as storms and rain. We then applied a data filter to remove data with poor data quality and contamination by clouds/aerosol conditions. This may introduce biases in our monthly sampling of the post-filtered measurements. On page 12 of the paper, we addressed the issue of spatial and temporal bias due to data filtering in the following paragraph, and have therefore not modified the paper in response to the reviewer's comment.

> **"Spatial and temporal bias due to data filtering.** CLARS-FTS samples the Los Angeles basin using its standard measurement sequence. However, as described in Wong et al. (2015), certain months of the year are more prone to cloud and aerosol interference in the Los Angeles basin. This may introduce biases in the monthly sampling of post-filtered data. To accurately estimate the LA basin value, we used the weighted average $XCH_{4(XS)}/XCO_{2(XS)}$ regression slope, as the statistical weight for each reflection point is based on the number of samples passing through the data quality filters. We also performed a bootstrap analysis to ensure that there is no sampling bias in the regression slopes (Efron and Tibshirani, 1993)."

4. Also, do you have the same amount of observations for each hour of the day (do you have biases linked to the hour you were able to collect measurements regarding clear sky conditions) ?

Response: We have looked into this. In general, the number of post-filtered observations did not have a strong diurnal bias. However, during certain months such as in June and September, our observations in the early morning and/or in the late afternoon were impacted by cloud or aerosol contaminations. A bootstrap analysis has been performed to ensure that there is no sampling bias in the regression analysis. This issue can be better addressed in the future using a 3D computer model to simulate the CLARS observations. To address this comment, we have added the sentence "The number of post-filtered observations did not have a strong diurnal bias however." at line 28 of page 12.

**Detailed comments:**

1. p.4 Line 23: the chosen strategy is, in each measurement cycle, to collect one set of LABS measurements and four SVO measurements. Please explain the motivation for this strategy.

Response: To address this, we have included the following sentence at line 26 on page 4, "Four SVO measurements are performed per measurement cycle so that any variability in the background during each measurement cycle, which typically lasts for 90 minutes, can be captured."

2. p.6 lines 5-7 : please choose one single notation (ppm CO2)-1 or ppm-1

Response: We changed the notation ppm-1 to (ppm CO2)-1. The sentence (now at line 9 of page 6) has been edited to "This is consistent with previous atmospheric observations: 7.8±0.8 ppb $CH_4$ $(ppm\ CO_2)^{-1}$ from TCCON in 2007-2008, 6.7±0.6 ppb $CH_4$ $(ppm\ CO_2)^{-1}$ from ARCTAS in 2008, and 6.7±0.0 ppb $CH_4$ $(ppm\ CO_2)^{-1}$ from CalNex in 2010 (Wunch et al., 2009; Wennberg et al., 2012; Peischl et al., 2013)."

3. p.6 lines 6-7 : please make it clearer : are the +/-0.8 and +/-0.0 indicating the uncertainty of the results or their variability ? How do the uncertainties compare between this study and the former ones ?

Response: They are reported as the statistical uncertainties of the regression slopes. The uncertainties of the regression slopes depend on both the variabilities and measurement uncertainties of the data. Figure 2 shows the variability of the CLARS-FTS monthly R values and uncertainties. The uncertainties in the CLARS-FTS R values are similar to previous studies. We have clarified this in the text (line 7 on page 6). This sentence has been further revised to, "During this period, R ranged from 5.4±0.4 ppb $CH_4$ $(ppm\ CO_2)^{-1}$ to 7.7±1.0 ppb $CH_4$ $(ppm\ CO_2)^{-1}$ with an overall mean and standard deviation of 6.5±0.5 ppb $CH_4$ $(ppm\ CO_2)^{-1}$. This is consistent with previous atmospheric observations and their uncertainties: 7.8±0.8 ppb $CH_4$ $(ppm\ CO_2)^{-1}$ from TCCON in 2007-2008, 6.7±0.6 ppb $CH_4$

(ppm $CO_2$)$^{-1}$ from ARCTAS in 2008, and 6.7±0.0 ppb $CH_4$ (ppm $CO_2$)$^{-1}$ from CalNex in 2010 (Wunch et al., 2009; Wennberg et al., 2012; Peischl et al., 2013)." We also added a footnote on the same page "*Peischl et al. (2013) reported 6.70±0.01 ppb $CH_4$ (ppm $CO_2$)$^{-1}$ from CalNex in 2010."

4. p.7 line 30 : The inventory-R based value underestimation of 30% seems effectively much larger than the CLARS R uncertainties, but please quantify this later (apparently from Fig.2, something like 3% ?).

Response: We believe that the reviewer would like us to quantify the CLARS-FTS R uncertainties in this sentence. We have edited the sentence to include the uncertainties, and revised the systematic difference between the inventory-R and CLARS R values. This sentence (at line 2 of page 8) is revised to "Figure 4 shows the annual R values determined from CLARS observations. CLARS annual R values were 6.4±0.1 ppb $CH_4$ (ppm $CO_2$)$^{-1}$, 6.2±0.1 ppb $CH_4$ (ppm $CO_2$)$^{-1}$, 6.5±0.1 ppb $CH_4$ (ppm $CO_2$)$^{-1}$, 6.5±0.1 ppb $CH_4$ (ppm $CO_2$)$^{-1}$ and 6.4±0.1 ppb $CH_4$ (ppm $CO_2$)$^{-1}$ in 2011, 2012, 2013, 2014 and 2015 respectively. The inventory-based R value systematically underestimated the observed annual R values by about 20 to 25% during the time period from 2011 to 2013."

5. p.8 lines 12-13 : You made the choice of distributing regularly the CARB CO2 emissions on the twelve months of the year. However, as you mention p.9 lines 25-26, the three better resolved inventory show similar monthly variability. Why don't you apply this variability around the mensual mean to distribute the annual CARB CO2 emissions? This would likely be more accurate and interesting to compare with the three highly resolved inventories.

Response: The reason we did not use the CARB $CO_2$ emissions in our calculation is because the official CARB emission inventory is an annual statewide (California) estimate. To estimate the monthly $CO_2$ emission for the basin from the CARB inventory, we have to first scale it to regional emissions and then apply the monthly variability from Hestia. Through these steps, we will introduce additional uncertainties in the derived emissions. Since Hestia provides monthly emission estimates for the South Coast Air Basin, we believe that Hestia is more accurate than the scaled CARB emissions. To address this comment, we added the following text "We did not use the CARB $CO_2$ emissions in our calculation because the official CARB emission inventories are annual statewide estimates. To derive the monthly $CO_2$ emissions for the basin from the CARB inventory, we have to first scale it to regional emissions by population and then apply the monthly variability from Hestia. Through these steps, we will introduce additional uncertainties in the derived emissions." at line 11 of page 10 of the updated manuscript.

6. p.10 line 1 : Please explain why you believe more in Hestia estimates than in the others.

Response: We have addressed the reason why we chose Hestia over CARB $CO_2$ emissions in our calculations. Please refer to our response to the previous comment. In addition to that, the following text at line 28 of page 9 explained why we chose Hestia over ODIAC and FFDAS, "As shown in Fig. 5, there are differences as large as 3 Tg $CO_2$ per month among the three gridded datasets: Hestia, ODIAC and FFDAS. The differences result from 1) emission calculation methods, 2) the underlying dataset used in the emission

calculations, and 3) spatial modeling. Hestia is derived primarily from local data in the South Coast Air Basin while ODIAC and FFDAS are based primarily on national and global proxy approaches. It has been shown that the use of a global dataset may underestimate emissions in Los Angeles by up to 18% (Brioude et al., 2013)."

7. K. Gurney is a co-author of the study, please remove (K. Gurney, 10 personal communication, 2016).

Response: Changes have been made in the text. The sentence (at line 21 on page 10) has been updated to "For $CO_2$ emissions, we assumed a 10% uncertainty in the Hestia monthly $CO_2$ emissions."

8. p.10 lines 19-20 : please quantify what Ân ́ partial Âz ̇ means here (see my specific comments).

Response: This sentence stated "Note that monthly variability in 2011 and 2015 was calculated on partial annual data". We believe the reviewer would like us to explain what "partial annual data" means in this sentence. We have clarified this in the sentence. This sentence (now at line 3 on page 11) has been revised to "Note that monthly variability in 2011 and 2015 was calculated on partial annual data, that is, from September to December in 2011 and from January to August in 2015."

9. p.13-15 : It would be interesting to give also here some information on the role of the different emission sectors as seen by the monthly-resolved inventories, and to compare this information with the top-down results cited in this section.

Response: This is a good point. We were hoping to do so. Unfortunately, there is no monthly-resolved inventories for us to compare with our top-down results now. There are only annual statewide total emissions at this point. We are aware that there are researchers working on this. We hope to be able to do some comparisons in the future when these data become available. To address this comment better, we have edited the sentence at line 22 of page 13 from "Currently, no monthly methane emission database is publicly available for comparison with our top-down estimates during our observational period." to "Currently, there is no monthly-resolved inventories available for us to compare with our top-down results. When these data become available in the future, we hope to understand better the role of each $CH_4$ source in the monthly variability we observed in total $CH_4$ emissions in Los Angeles." on page 13, line 22.